# ConMe: Rethinking Evaluation of Compositional Reasoning for Modern VLMs

**Irene Huang**[*1]    **Wei Lin**[*2]    **M. Jehanzeb Mirza**[*†1]    **Jacob A. Hansen**[1]
**Sivan Doveh**[3]    **Victor Ion Butoi**[1]    **Roei Herzig**[4]    **Assaf Arbelle**[3]
**Hilde Kuehne**[5,6]    **Trevor Darrell**[4]    **Chuang Gan**[5,7]    **Aude Oliva**[1,5]
**Rogerio Feris**[5]    **Leonid Karlinsky**[5]

[*]Equally Contributing Authors.
[1]MIT, USA.    [2]JKU, Austria.    [3]IBM Research, Israel.    [4]UC Berkeley, USA.
[5]MIT-IBM, USA.    [6]Tuebingen AI Center, Germany.    [7]UMass Amherst, USA.

Dataset: https://huggingface.co/conme/ConMe
Code: https://github.com/jmiemirza/ConMe

## Abstract

Compositional Reasoning (CR) entails grasping the significance of attributes, relations, and word order. Recent Vision-Language Models (VLMs), comprising a visual encoder and a Large Language Model (LLM) decoder, have demonstrated remarkable proficiency in such reasoning tasks. This prompts a crucial question: have VLMs effectively tackled the CR challenge? We conjecture that existing CR benchmarks may not adequately push the boundaries of modern VLMs due to the reliance on an *LLM only* negative text generation pipeline. Consequently, the negatives produced either appear as outliers from the natural language distribution learned by VLMs' LLM decoders or as improbable within the corresponding image context. To address these limitations, we introduce ConMe[1] – a compositional reasoning benchmark and a novel data generation pipeline leveraging VLMs to produce 'hard CR Q&A'. Through a new concept of VLMs conversing with each other to collaboratively expose their weaknesses, our pipeline autonomously generates, evaluates, and selects challenging compositional reasoning questions, establishing a robust CR benchmark, also subsequently validated manually. Our benchmark provokes a noteworthy, up to 33%, decrease in CR performance compared to preceding benchmarks, reinstating the CR challenge even for state-of-the-art VLMs.

## 1  Introduction

Present day Vision-Language Models (VLMs) [1–12] have recently emerged as the default choice for many computer vision tasks. However, these models also have their Achilles' heel. Several recent studies have highlighted important VLM failure modes, especially their lacking ability to perform Compositional Reasoning (CR) [13–17]. CR is the ability of the VLM to recognize and attend to the language concepts beyond objects (*i.e.,* nouns), such as attributes, relations, fine-grained object alternatives, and more, in both the image and text of a VL pair. As noted in [13–17], earlier dual-encoder VLMs (*e.g.,* CLIP [1]) have especially low, even close to chance, CR performance. However, more modern VLMs, which combine a pre-trained vision encoder with a strong LLM decoder (*e.g.,* LLaVA [10]) and employ both architectural (projection layer/MLP, tuning

---

[†]The work was done while being at TU Graz, Austria. Correspondence: `jmirza@mit.edu`
[1]ConMe is an abbreviation for 'Confuse Me'.

38th Conference on Neural Information Processing Systems (NeurIPS 2024) Track on Datasets and Benchmarks.

of the LLM decoder) and instruction-tuning-based alignment [18, 19], demonstrate much stronger performance on compositional reasoning task when evaluated on present CR benchmarks [14–16].

Most CR benchmarks [14–16] have been formed from collections of text-image pairs, grouped by the presence of certain CR concepts, such as relations, attributes, *e.t.c.*, by a process of randomly "flipping" the present CR concept in the positive text to form a "negative alternative" text (having the CR concept wrong). The VLM's preference for the resulting negative is then compared to the true positive source text thus testing the VLM's ability to entail the correct text from the image.

Originally, simple word substitution or ordering changes were used for this CR concept flipping [14–16], relying on simple language augmentation heuristics and tools. This simple approach was able to elegantly illustrate the CR fail modes of dual-encoder VLMs [1, 20, 21]. This can be intuitively explained due to their contrastive pretraining, for which representing only the objects (nouns) is sufficient to disambiguate all the text-image pairs in a random batch of limited size. However, modern VLMs, *e.g.,* [19], demonstrate significantly higher performance on such benchmarks. We conjecture that their increased CR performance stems from two factors: (i) the negative synthesis heuristic may generate "out-of-natural-language-distribution" samples and is not powerful enough to "fool" the LLM decoders of the VLMs; (ii) even if the language of the produced negative is in-distribution, the produced CR concept manipulation that forms the negative text might be unlikely for the scene observed in the corresponding image. As we observe in our evaluations in Section 4.2, even the SugarCrepe benchmark [17] designed specifically to generate in-language-distribution samples (to thus not suffer from factor

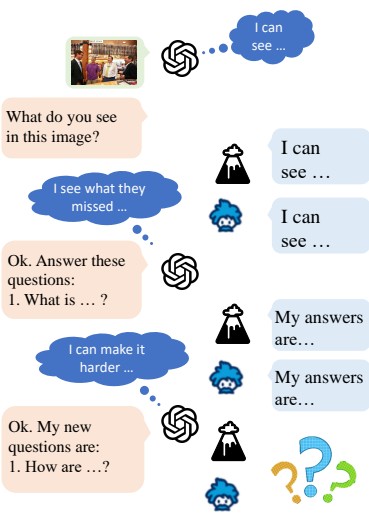

Figure 1: We propose a new concept of VLMs conversing with each other to collaboratively expose their weaknesses. Our pipeline autonomously generates, evaluates, and selects challenging Compositional Reasoning (CR) questions, to establish a robust CR benchmark – **ConMe**.

(i)) by using LLMs for negative synthesis and applying language-side debiasing, likely suffers from factor (ii), as not looking at the image can produce unlikely negatives (w.r.t. the image). This naturally leads us to ask – did the modern VLMs relying on LLM decoders solve the previous issue with the low CR performance of dual-encoder VLMs?

We propose a new CR benchmark ConMe, generated through our novel automated data generation pipeline utilizing GPT-4V [22] with a combination of open-source modern VLMs to answer this question in the negative. Our pipeline gradually discovers what the stronger VLM (GPT-4V) sees/knows and other (evaluated) VLMs do not, to produce plausible and difficult question-and-answer options for CR testing. In a way, our pipeline creates a 'conversation' among VLM agents to collaboratively expose their weaknesses to GPT-4V, as shown in Figure 1. Using VLMs in the pipeline instead of relying on LLM generation, we propose a new method of incorporating both image and language context during the benchmark curation, thus avoiding suffering from factors (i) or (ii) as mentioned above. Finally, we show that our conclusions on hardness generalize to strong unseen VLMs likely due to the similar nature of their Visual Instruction tuning alignment methodology.

To summarize, our contributions are as follows: (i) Through extensive experiments we show that compositional reasoning is still a significant problem for present-day VLMs and their CR performance can be evaluated more closely than currently possible by existing CR benchmarks; (ii) We propose a novel CR data generation pipeline that incorporates GPT-4V and contemporary open-source VLMs which can potentially be used to generate abundant challenging CR data; and (iii) We also contribute a challenging CR benchmark ConMe, which leads to up to 33% decrease in CR performance of SOTA VLMs as compared to the preceding benchmark. Moreover, we accompany ConMe with an LLM-based analysis tool for automatic mining of insights on VLMs' CR weaknesses.

## 2   Background and Related Work

**Vision-Lanugage Models.** There has been a remarkable surge in VLMs, which have shown impressive performance for many vision-language understanding tasks, *e.g.,* zero-shot classification, visual question-answering (VQA), image captioning, *e.t.c*. In a broader sense, the present day VLMs can be

divided into two families. One family of methods relies on dual-encoders (vision and text encoder) and usually trains the encoders with a contrastive objective by using a large corpus of paired image-text data scraped from the web. Most common among these methods are CLIP [1], ALIGN [23] and OpenCLIP [24]. Many different ideas have been explored to improve these models, *e.g.,* by using off-the-shelf object detectors [25–27], using cross-attention and additional regularization objectives [28–31], filtering noisy captions (*e.g.,* BLIP [4]), employing textual nearest-neighbors [2], using geometrically consistent representations [21], caption augmentations [32, 33]. In parallel, some other methods employ few-shot supervision [34–36] and label-free finetuning [37–40]. The other family of methods aligns the visual modality with a frozen LLM. BLIP-2 [41] bridges the modality gap between a pre-trained visual encoder and an LLM by using a Querying Transformer. InstructBLIP [19] proposes to improve [41] by employing instruction tuning. MiniGPT [9] grounds a vision encoder with a frozen LLM (Vicuna [42]) by only using a (trainable) linear projection layer between the two. MiniGPT-V2 [11] replaces the LLM with LLaMA-2 [43] and also proposes to unfreeze it during the training and finetuning phases. LLaVA [44] also grounds an LLM with a pre-trained visual encoder and proposes Visual Instruction Tuning by carefully curating instruction-response pairs, to enhance the performance, with further improvements proposed in [45]. Some other works [44, 46–51] also explore similar ideas and propose certain improvements.

**Compositional Reasoning Benchmarks.** Compositionality Reasoning is the VLM's ability to attend to more complex concepts in natural language, beyond only objects (*i.e.,* nouns). Recently, many benchmarks have been proposed to evaluate CR in VLMs. Winoground [13] proposes a simple task to evaluate the CR ability of VLMs – given two images and captions, the goal is to correctly match them, where the captions contain the same words but in different orders. They show that the SOTA VLMs only perform slightly better than chance. COLA [52] proposes a benchmark where the task for the VLM is to retrieve correct images (based on the correct configuration of attributes and objects) from a database, where *distractor* images are present. Crepe [16] also evaluates several SOTA VLMs and highlights the lack of CR abilities. The evaluation is performed on a proposed benchmark inspired by cognitive science literature, specifically testing for systematicity and productivity. Attribution, Relation, and Order (ARO) benchmark [15] is a large-scale dataset specifically designed to evaluate the VLM's ability to understand the relational and order abilities. SugarCrepe [17] shows that the current benchmarks proposed to test the VLM's ability for compositionality are easily *hackable*. They evaluate *blind* models having no access to image data and show that they outperform modern VLMs on these benchmarks. To fix this, they forego the rule-based hard negative generation by employing LLMs and propose adversarial refinement. However, they mostly focus on "syntactic correctness" and language model-based hackability and de-biasing, also without checking VLM task hardness. As a result (as demonstrated in Section 4.2) modern VLMs (employing strong LLM decoders) experience no significant challenge on SugarCrepe [17], even compared to Crepe, its predecessor. To counter these shortcomings in the previous benchmarks, we propose ConMe which is generated through an automated pipeline that foregoes relying only on the LLMs by employing VLMs to generate new CR question-and-answer options by leveraging a multi-turn conversation between stronger (GPT-4V) and weaker VLMs to increase question difficulty by gradually uncovering what weaker models are blind to, as well as incorporating image context into the hard negative formation.

**V&L Benchmarks QA Analysis Taxonomies** Taxonomies, if available, in V&L Benchmarks are acquired either by manual verification [53–55] or from the existing annotations of the image source [15–17]. In our proposed ConMe curation, a group of VLMs (including GPT-4V) communicate with each other collaboratively uncovering their CR weaknesses, such that GPT-4V is able to generate CR QA that are difficult for all VLMs (including unseen) and GPT-4V itself. The available VLM error taxonomies related to our generated CR QA in ConMe are therefore dynamic, scalable, and adaptive. This necessitates an automatic LLM-based taxonomy analysis tool for generating interesting insights using our ConMe benchmark. We contribute such a tool in this work (Section 5.2).

## 3   ConMe: A Compositional Reasoning Benchmark

In contrast to the previous benchmarks [16, 17] which are created either by rule-based manipulation of the text to create the negatives or use LLMs for this task, our ConMe is harnessed by employing additional image context using 'a conversation' between state-of-the-art VLMs. Our negative text generation pipeline can create abundant challenging CR data, given only a random collection of images. In this paper, we use our proposed text generation pipeline on the images present in the SugarCrepe [17] dataset and come up with a challenging CR benchmark labeled ConMe. In the

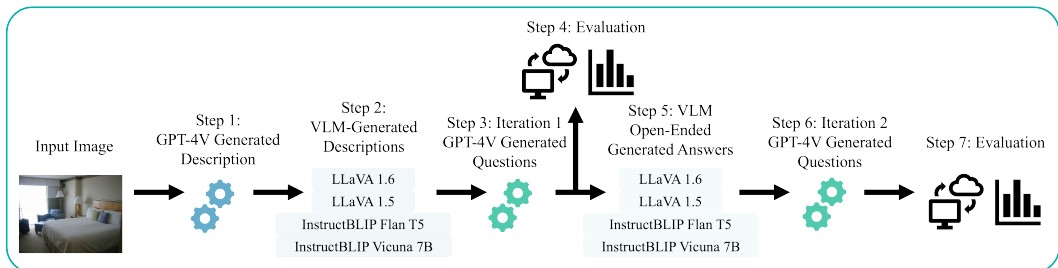

Figure 2: Our proposed CR data generation framework employs multiple open VLMs in a multi-stage 'conversation' setup. Given an image, first, GPT-4V and the VLMs are prompted to describe the image in detail. Then, providing all the generated descriptions from the VLMs and the GPT-4V itself as context, GPT-4V is tasked with the generation of the first iteration of CR questions, and the VLMs are evaluated on these questions and also prompted to generate open-ended answers. Finally, GPT-4V is again employed and prompted to generate *more* challenging CR questions with the additional context from the previous iterations output resulting in challenging CR questions, and their correct answers (positives) and confusing wrong answers (negatives).

following, we provide details about our proposed automated CR data generation pipeline. A brief overview of the SugarCrepe dataset partitions is provided below and in the Appendix Section E.

## 3.1 Hard Compositional Reasoning (CR) QA Generation Pipeline

Our proposed hard negative mining pipeline consists of GPT-4V and several open VLMs (LLaVA 1.6-7b [56], LLaVA 1.5-7b [12], InstructBLIP Flan-T5 [18], and InstructBLIP Vicuna-7b [42]) deployed in a multi-stage setup. An overview of all the stages is provided in Figure 2 and next we describe them in detail. More details and the full prompts used are provided in the Appendix Section A.

**GPT-4V Generated Description (Stage 1)** In the first stage, we prompt the GPT-4V to generate a detailed description of the input image. We treat this as the "ground truth" description, to capture as many details that the model deems itself confident in describing.

**Downstream VLM Generated Descriptions (Stage 2)** In stage 2, we prompt each targeted downstream VLM to generate a detailed description for the same input image, by similarly prompting the VLMs as in stage 1. This discloses to GPT-4V what the open VLMs 'see' (or 'pay attention to') on this input image, which is helpful for comparison in the next stages.

**Iteration 1 of GPT-4V Generated Questions (Stage 3)** In this stage, we provide the GPT-4V with the descriptions that it generated (in stage 1), along with descriptions from the individual downstream VLMs, and prompt it to generate multiple (*e.g.,* 10) challenging CR questions based on these generated descriptions. Furthermore, we also prompt the model to provide an answer (positive) and multiple corresponding negatives to that positive. The goal here is to provide GPT-4V with the descriptions that the downstream VLMs generated so that it can come up with targeted reasoning questions about the details that the open-source VLMs missed.

**VLM Inference Evaluation to Iteration 1 Questions (Stage 4)** Each CR question-answer from stage 3 is now framed as a binary multiple-choice selection between two answer options, one of which is the correct answer and the other is the negative option. We employ the 'generate' evaluation mode (asking the VLM to generate the index of the correct choice) to evaluate each of the four downstream VLMs and record the resulting accuracies for each model. Later, we perform an intermediate filtering step. Specifically, for each collected data sample (*i.e.,* input image, question, correct answer, and one negative option), we evaluate all the downstream VLMs (*i.e.,* LLaVA 1.6-7b, LLaVA 1.5-7b, InstructBLIP Flan-T5, InstructBLIP Vicuna-7b). If all four VLMs answered correctly, we discard this data sample; conversely, if at least one VLM answered incorrectly, we keep the sample. After filtering, we obtain the updated accuracy values for each of the four models.

**Open-Ended Answer Generation to Iteration 1 Questions (Stage 5)** Given the first iteration of generated CR questions, we pass these to the four open VLMs to obtain their open-ended answers intending to utilize the resulting responses to provide GPT-4V with additional context on what image details the models do not perceive (failing on).

| Benchmarks | replace-att | replace-obj | replace-rel | Total |
|---|---|---|---|---|
| SugarCrepe | 788 | 1652 | 1406 | 3846 |
| ConMe | 8863 | 8691 | 6793 | 24347 |

Table 1: Total number of samples per partition in the SugarCrepe and our ConMe benchmarks.

**Iteration 2 of GPT-4V Generated Questions (Stage 6)** In this stage, we again prompt the GPT-4V with additional context asking it to generate more challenging CR data. Specifically, we prompt the GPT-4V with the set of 10 CR questions it generated (*cf*. stage 3) and also the open-ended answers from the open VLMs and instruct it to generate more challenging CR questions. The intuition behind prompting the model in this manner is to make GPT-4V reflect upon its own generated data and also have the additional context of the details, the downstream VLMs focused upon, in their answers.

**VLM Inference Evaluation to Iteration 2 Questions (Stage 7)** This stage is concerned with the final evaluation performed on the second iteration of GPT-4V-generated questions. Similarly to stage 4, we again use the 'generate' inference evaluation with the questions framed as a binary multiple-choice selection. Also similar to stage 4, we again filter out any samples that all four VLMs answer correctly. Accordingly, we record the evaluation results for the resulting filtered dataset (ConMe) – the final curated dataset produced by our pipeline.

Our proposed pipeline curates a new dataset for a set of input images from the 3 SugarCrepe [17] partitions, which consists of 919 total images[2] – 333 from the *Replace-Att* partition, 333 from *Replace-Object*, 253 from *Replace-Relation*. However, our automated hard CR QA pipeline can generate challenging compositional reasoning QA on an arbitrary set of images. Extensive experimentation in Section 4 highlights the challenging nature of our proposed ConMe dataset which also generalizes beyond the set of 4 open VLMs employed during ConMe curation.

## 4 Experiments

We first provide implementation details we employ for our ConMe curation pipeline, then introduce the dataset partitions and later discuss the results of 7 VLMs (including GPT-4V itself) on our contributed ConMe dataset.

**Implementation Details** We employ GPT-4V [22] through the OpenAI API, using the gpt-4-vision-preview endpoint. We use the default setting for the image resolution and limit the number of new tokens generated by the model to 2000. We use four open VLMs for ConMe curation: LLaVA 1.6-7b [56], LLaVA 1.5-7b [12], InstructBLIP Flan-T5 [18], and InstructBLIP Vicuna-7b [42]. For open-ended text generation, for the two LLaVA models, we use temperature 0 and max new tokens 500; for the two InstructBLIP models, we use temperature 1.0, max new tokens 500, top 90% probability mass, repetition penalty 1.5, length penalty 1.0, and number of beams 5. When prompting the VLMs to answer binary multiple choice questions, for both LLaVA models, we use temperature 0 and max new tokens 128; for both InstructBLIP models, we use temperature 1.0, max new tokens 10, top 90% probability mass, repetition penalty 1.0, length penalty 1.0, and number of beams 5. The above settings are directly taken from the respective publications and empirically validated for best performance. For a fair comparison, for generation inference mode when prompting these VLMs to answer binary multiple-choice questions, we use the same generation parameters as those we used for evaluating the InstructBLIP models. Furthermore, we also validate that the same conclusions are obtained by switching to 'perplexity' inference for multiple choice questions (choosing the answer by minimal CLM loss value computed by the LLM decoder [58, 59]).

### 4.1 Datasets

The three SugarCrepe partitions are structured to target a particular CR aspect (*i.e.,* attributes, objects, or relations) within an image, and provide only a single question per aspect. On the other hand, to construct ConMe, we generate various CR questions for each image, thus resulting in a larger dataset in terms of sample count. The total sample size for the SugarCrepe and ConMe datasets is listed in Table 1. Thanks to our proposed automated ConMe curation pipeline, this dataset can be further expanded by incorporating an arbitrary image set.

---

[2]sourced from the MS-COCO [57] validation set

| Models | Seen | SugarCrepe | ConMe | Manual Subset | Performance Drop |
|---|:---:|:---:|:---:|:---:|:---:|
| LLaVA 1.5-7b | ✓ | 88.5 | 57.7 | 56.2 | -30.8 |
| LLaVA 1.6-7b | ✓ | 89.2 | 57.5 | 54.9 | -31.7 |
| InstructBLIP Flan-T5 | ✓ | 91.4 | 58.5 | 59.8 | -33.0 |
| InstructBLIP Vicuna-7b | ✓ | 82.5 | 53.6 | 46.2 | -28.9 |
| InternLM-XComposer2-VL-7b | ✗ | 92.0 | 79.7 | 82.1 | -12.3 |
| Idefics2-8b | ✗ | 85.5 | 70.1 | 72.1 | -15.4 |
| GPT-4V | | 91.2 | 80.1 | 81.8 | -11.2 |
| Mean | | 88.6 | 65.3 | 64.8 | -23.3 |

Table 2: Average accuracy (%) across baseline and our generated data partitions using generate inference evaluation mode. **ConMe** results refer to the evaluation of all the data samples, while **Manual Subset** results are obtained from a manually verified subset of 1000 samples.

## 4.2 Results

We evaluate 6 strong open VLMs – the four used in ConMe curation and two 'unseen' models, on the baseline SugarCrepe dataset and the ConMe dataset curated through our proposed CR QA generation pipeline (Sec. 3). The 'unseen open VLMs': InternLM-XComposer2-VL-7b [60] and Idefics2-8b – recent improvement of Idefics1 [61], were used to evaluate the generalization of ConMe to models not seen during its curation. Furthermore, we also evaluate GPT-4V itself on ConMe showing its performance significantly decreases compared to the original SugarCrepe. Surprisingly, using our proposed ConMe pipeline that employs GPT-4V for hard CR QA generation - GPT-4V is shown to 'fool' itself! For comparison between the baseline and pipeline datasets, we calculate the evaluation accuracy for all 3 partitions of the original SugarCrepe and average the 3 numbers into the final metric reported in Table 2. We observe a substantial performance drop of 23.3% from original SugarCrepe when averaged over the 7 evaluated VLMs. When comparing the results for the 4 'seen' (during ConMe curation) models on the baseline SugarCrepe dataset and our curated ConMe benchmark, we see even more significant performance drops as expected. For example, we observe a performance drop of up to 31.7% and 33.0% on the LLaVA and InstructBLIP families of models, when evaluated on the more challenging ConMe benchmark as compared to the original SugarCrepe.

Since our ConMe curation pipeline is employing VLMs (that 'converse' with GPT-4V), it is also important to evaluate our ConMe benchmark on unseen VLMs to test its generalization ability. For this purpose, we evaluate 2 state-of-the-art open VLMs: InterLM-XComposer, Idefics2. And we also report results for GPT-4V itself as it was not targeted for the hard CR QA production (rather, it generated them). We observe from the results in Table 2 that our ConMe also generalizes to unseen VLMs and is even challenging for GPT-4V which is often considered as the strongest VLM currently available. We see that for unseen VLMs our ConMe benchmark provokes a performance drop of up to 15.4%. Notably, we also observe a performance drop of 11.2% when evaluating GPT-4V. These results show that our proposed ConMe benchmark is not only challenging for the VLMs employed in our CR QA generation pipeline but can also generalize to unseen SOTA VLMs.

**Manually Verified ConMe Partition.** Our CR QA generation pipeline employs a conversation between multiple generative models, thus it can be prone to issues like hallucinations in text, resulting in unfair model evaluations. To analyze such issues, we manually verified a subset of 1000 samples[3] from the ConMe dataset and also reported the accuracy on this subset in Table 2. Manually verified partition is contributed as part of ConMe. Comparing the evaluation on the entire ConMe vs its manually-verified partition (Tab. 2), the performance is almost the same, even with the manual-verified partition being 0.5% 'harder' on average. These results show that our CR QA generation pipeline is robust to common generative-models-based errors and (automatically curated) ConMe predictions, in terms of CR strengths and weaknesses analysis of seen and unseen VLMs, are trustworthy. We attribute this to the effective multi-stage filtering performed in our generation pipeline.

---

[3]The subset resulted from reviewing ConMe CR QA samples in random order until 1000 human-validated samples were gathered. We found $\sim 20\%$ errors during manual verification. However, while verifying, we observed that the mistakes found were uniformly distributed and largely independent from mistakes made by the VLMs which results in negligible performance fluctuation between performance predicted by (unfiltered) ConMe pipeline data and 1000 Human validated CR QA from ConMe, as reported in Table 2.

| Model | Partition | Iteration 1 | | Iteration 1 + Filtering | | Iteration 2 | | Iteration 2 + Filtering | |
|---|---|---|---|---|---|---|---|---|---|
| | | Perplexity | Generate | Perplexity | Generate | Perplexity | Generate | Perplexity | Generate |
| LLaVA 1.6-7b | replace-att | 82.2 | 82.2 | 69.0 | 64.7 | 79.3 | 79.2 | 56.5 | 57.9 |
| | replace-obj | 83.5 | 83.2 | 70.7 | 65.7 | 79.9 | 80.1 | 51.6 | 57.0 |
| | replace-rel | 82.9 | 82.9 | 70.5 | 65.6 | 79.7 | 79.5 | 57.8 | 57.6 |
| LLaVA 1.5-7b | replace-att | 79.5 | 79.5 | 64.3 | 59.2 | 73.5 | 73.8 | 58.2 | 58.5 |
| | replace-obj | 80.4 | 80.7 | 65.1 | 60.4 | 73.9 | 73.6 | 51.3 | 56.8 |
| | replace-rel | 79.6 | 80.0 | 64.8 | 59.6 | 73.6 | 73.4 | 56.5 | 57.8 |
| InstructBLIP Flan-T5 | replace-att | 78.7 | 78.1 | 62.8 | 56.3 | 76.4 | 77.1 | 62.1 | 59.8 |
| | replace-obj | 79.3 | 78.5 | 63.1 | 56.1 | 76.0 | 75.9 | 61.1 | 57.4 |
| | replace-rel | 78.4 | 78.0 | 62.6 | 55.6 | 75.8 | 75.7 | 58.7 | 58.2 |
| InstructBLIP Vicuna-7b | replace-att | 61.0 | 69.4 | 31.9 | 39.1 | 58.3 | 64.4 | 45.9 | 52.1 |
| | replace-obj | 61.5 | 70.2 | 31.5 | 38.9 | 58.8 | 66.8 | 47.4 | 54.7 |
| | replace-rel | 60.6 | 70.1 | 32.0 | 39.6 | 58.7 | 65.6 | 56.1 | 54.0 |

Table 3: Accuracy (%) using perplexity and generate inference evaluation modes while comparing performance across the generated partitions for iterations 1 and 2 of GPT-4V generated CR questions.

## 5 Analysis and Ablations

This section provides detailed ablations and analysis of our proposed benchmark. We first ablate the effect of multiple stages of filtering performed in our ConMe curation framework, then compare the generation and perplexity-based inference method found in the literature, and finally provide insights into the error taxonomy contributed as part of ConMe and automatically applied to analyze all the evaluated VLMs drawing interesting conclusions and insights on their strength and weaknesses. For completeness, we also delegate some qualitative examples to the Appendix Section D.

### 5.1 Comparison with Perplexity Inference Evaluation

For modern VLMs, two different types of evaluation methods are used by the community. Namely, the generation and the perplexity:

**Generation:** For generation evaluation, we use the forward call of the model to output a letter option, A or B, and we record sample accuracy based on this generated output letter, after string-comparing it with the ground truth. The main results in Table 2 are obtained by evaluating the models in the generate mode.

**Perplexity:** For perplexity evaluation, we calculate the loss score (perplexity score) associated with each answer option, and we determine the sample accuracy by selecting the letter associated with the smaller loss value [58, 59]. More formally, denoting the VLM visual encoder by $\mathcal{E}_V$[4] and the LLM decoder by $\mathcal{D}_L$ the perplexity score $P(I, T)$ for an image $I$ and a text $T$ ($T$ can include a prompt prefix) is defined as:

$$-log\mathcal{P}(T|I) = -log\left(\frac{1}{|T|}\sum_{i=1}^{|T|}\mathcal{L}(\mathcal{D}_L(\mathcal{E}_V(I), T_{[1:i-1]}), T_i)\right),\qquad(1)$$

For completeness, the prompt templates used for perplexity inference, as some validation evidence supporting the prompts used are provided in the Appendix Section B.

In Table 3 we provide the detailed results with both the evaluation protocols on the three partitions from our curated ConMe benchmark. We see that both evaluation protocols provide consistent results for the different partitions. Furthermore, we also see the effect of multi-stage filtering proposed in our ConMe curation framework. With each successive filtering step, the benchmark becomes more challenging. For example, the accuracy drops by $\sim 30\%$ on average (after stage 2 filtering) for the four models as compared to the average accuracy obtained in the first step of the framework. Moreover, the difference between the results obtained from the perplexity and generate inference mode is only $\sim 2\%$ on average, signifying the consistency of the obtained results regardless of the inference approach. We also ablate the baseline SugarCrepe dataset for these two evaluation modes and also for the VQAScore [62] and provide the results in the Appendix Section B.

---

[4]For ease of notation in case of [12], $\mathcal{E}_V$ will include the projection MLP, and in case of [19], $\mathcal{E}_V$ will include the Q-former

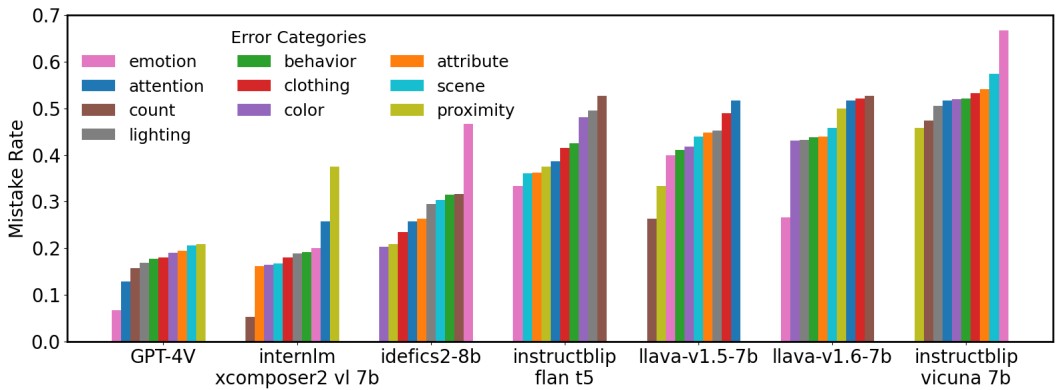

Figure 3: Distribution of mistake rates of various VLMs across different error categories automatically obtained by our proposed analysis framework. Table 6 in the Appendix specifies each error category.

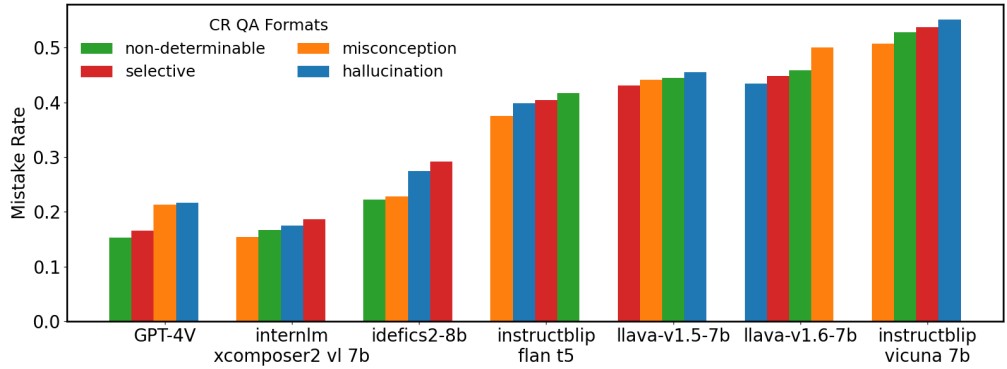

Figure 4: Distribution of mistake rates of various VLMs across different CR QA formats automatically obtained by our proposed analysis framework. Tab. 5 in the Appendix specifies each CR QA format.

## 5.2 Error Taxonomy Analysis

We complement the ConMe curation pipeline – our automatic framework for mining hard CR QA, by also contributing an LLM-based analysis tool for automatic categorization of VLM mistakes according to human-specified error taxonomies provided simply by natural language description. This analysis pipeline is necessary, as our ConMe curation pipeline is automatic, adaptive (to the target VLMs being analyzed), and scalable (in a sense we can scale arbitrarily by providing the ConMe curation framework with more images). Hence, the generated CR QA in ConMe need to be analyzed (categorized) automatically in order to dynamically mine insights on the evaluated VLM weaknesses in terms of the relative distribution of their errors across error categories or other CR QA insights specified by the taxonomies. For our analysis tool, we utilized LLaMa-3 8B [63] to categorize our human-filtered ConMe partition. We leverage two natural language taxonomy specifications provided in Tables 5 and 6 in the Appendix. The complete LLaMa-3 8B prompts are provided in Appendix Section C. In Figure 3 we plot the mistake rate for the 7 VLMs which are partitioned by the *error category type*. Similarly, Figure 4 provides the breakdown of VLM mistakes according to *CR QA formats*. We observe notable actionable insights (for future work) and interesting conclusions for different models. For example, LLaVA 1.6-7B showed a large improvement in 'emotion understanding' compared to LLaVA 1.5-7B (decreasing emotion error from 40% to 27%), but suffered a large decrease in 'counting ability' (counting mistakes rose from 26% to 53%). We also find that InternLM XComposer2 VL 7B struggles with proximity assessment, Idefics2-8B with emotion recognition, and InstructBLIP Flan T5 with counting. Interestingly, GPT-4V CR errors are more evenly distributed among error categories, peaking at proximity assessment errors, while our analysis also identified it's somewhat higher tendency to misconception and hallucination in terms of CR QA formats taxonomy. We delegate further error analysis to the Appendix Section C but in summary, our findings can lead to actionable improvement targets for each model. For example,

LLaVA 1.6 can benefit from more instructional data targeting improving the capability of fine detail analysis (like object attention and counting) while avoiding misconception or hallucination; Idefics2-8B could enhance higher-level reasoning such as emotion recognition, leveraging additional datasets in that area; InternLM XComposer2 VL 7B could benefit from additional training data focused on proximity detection. Such insights, instrumented by our ConMe and its analysis tool, are crucial for guiding future developments in VLM architectures, instruction data collection, and training methodologies, ensuring more robust compositional reasoning capabilities across diverse visual and textual contexts.

## 6 Conclusions and Limitations

We have presented a fully automated hard negative generation framework and a curated dataset ConMe for evaluating and analyzing CR performance of modern VLMs, which include an LLM decoder component and hence are more sensitive to language mistakes and learn to better interpret the provided image context. With thorough evaluation and analysis, we have found that our proposed approach is significantly more effective in detecting and targeting compositional reasoning failure modes of the state-of-the-art VLMs. Our work provides a pathway to building increasingly difficult (and adaptive to VLM evolution) benchmarks for even modern VLMs, as our proposed methodology can easily be employed to generate challenging CR data sources, given any arbitrary image collection. In the future, the proposed negative generation pipeline can also be extended to curate large-scale training datasets for finetuning models to improve the compositional reasoning aspects of these models, in addition to further analysis of the types of failure modes most common to modern VLMs.

**Limitations** Like any other research, our work also comes with certain limitations. The quality and robustness of the curated dataset rely strongly on the VLM used to generate the CR questions. In this regard, we selected GPT-4V for its demonstrated success in generative capabilities when processing both image and text inputs concurrently. Nevertheless, it is not perfect, and our proposed generation framework can introduce errors. However, the evaluations on the human-verified subset hint that the errors introduced in our ConMe dataset are uniformly distributed and the difference (in accuracy) as compared to the entire ConMe dataset is almost negligible. In the future, as the models become better, our CR data generation pipeline will also directly benefit. Furthermore, another way to address this limitation could also target analyzing the images associated with manually verified errors, to capture aspects of images that could be best used for applying the proposed pipeline to other image data collections.

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

# Appendix

In the appendix, we provide additional material helpful for understanding the main manuscript. Specifically, in Section A we list the prompts for the VLMs used in our QA generation pipeline, Section B provides the prompts used for perplexity-based evaluations, then Section C offers more details about the error taxonomy analysis. Finally, we conclude by providing some qualitative examples in Section D.

## A  Prompts for CR QA Generation

In this section, we provide the prompt templates used in the 7 stages of our pipeline for the generation of challenging CR QA. These are listed in Figures 5, 6, 7, 8, 9, 10 and 11.

## B  Perplexity Prompts and Evaluations

Here, we provide the complete prompt templates used to perform perplexity-based inference. These are listed in Figures 12 and 13. Furthermore, we also provide the results on the baseline SugarCrepe partitions in Table 4.

## C  Error Taxonomies Definition

Our automated CR QA generation pipeline can also be employed to find common failure modes in the VLMs. In the main manuscript, we discover several such taxonomies and analyze present-day VLMs for their common failure modes. For completeness, in Table 5 and 6 we provide comprehensive definitions of the discovered error categories. We also provide the prompt to Llama-3 in Figure 14 and the sample count in different error categories in Figure 15.

## D  Qualitative Examples

In Figure 16 we provide qualitative examples to focus on:

- Qualitative differences between the original image-caption matching task including their positive/negative captions and the GPT-4V generated question/answer options.
- Qualitative examples of question types generated using GPT-4V.

We observe that the GPT-4V generated questions can be more complex in language and focus, such as combining multiple compositional reasoning aspects (*e.g.,* first row, combining object count with relative location), noticing more fine-grained details such as aspects of lighting (*e.g.,* second row, shadows cast by sunlight), or locating more specific localization within an image (*e.g.,* third row, specific details about a smaller region rather than larger objects or the whole image at large). The finer details in the positive-negative pairs and more challenging questions present in our ConMe benchmark can be attributed to using visual data as a prior, instead of only relying on LLMs to curate the positive-negative pairs, as done by prior works [16, 17].

## E  SugarCrepe Partitions

Our ConMe benchmark utilizes the partitions provided by SugarCrepe [17] dataset, which consists of 919 total images[5] – 333 from the *Replace-Att* partition, 333 from *Replace-Object*, 253 from *Replace-Relation*. SugarCrepe proposes to modify the positive caption of an image by either replacing, swapping, or adding atomic concepts – which are demonstrated through different dataset partitions – in order to confuse the VLMs. To avoid language errors, SugarCrepe employs an LLM for the atomic concept manipulation and follows the manipulation by LLM-based de-biasing (ensuring that the LLM has no bias towards the augmented or the original text), yet only on the text side, disregarding the image context. On the contrary, in our work, we focus on providing image context in addition to

---

[5]sourced from the MS-COCO [57] validation set

textual context, by employing a combination of different VLMs, rather than LLMs, to generate new questions and answer options.

Below we include a summary and description of these three partitions from the baseline SugarCrepe dataset, to provide additional context on the original structure:

- *Replace-Attribute* forms a negative by replacing the attributes describing object characteristics. As an example, for an image taken on the ground, two text options are: {Several vehicles providing ground transportation are shown in the photo: streetcar, tour bus, classic car, and family cars.} and {Several vehicles providing aerial transportation are shown in the photo: helicopter, hot air balloon, small plane, and glider.}. We observe, that the negative was generated by the LLM without any image context. Hence, despite the linguistic correctness, it is unlikely a hard negative for a VLM provided with the image context of a ground.

- *Replace-Object* refers to negative generation via replacing the object (noun) in the positive caption. For example, given an image of a teddy bear next to some boxes in a room, a VLM is asked to choose between the *positive* {A big teddy bear sitting next to some boxes.} and the *negative* {A big car sitting next to some boxes.}. Even though the negative is grammatically correct and potentially unbiased given the partial context (a room is not mentioned in the positive text), we would not expect a car to sit next to the boxes in a room (though it might happen near the side of the road). As follows, it is unlikely that a modern VLM would be confused, as it can complete the missing details (the room) from the image and infer the unlikelihood of a car there based on the image context.

- *Replace-Relation* replaces a word describing a spatial relation between objects in a caption to form the negative. For example, given an image taken in a bedroom, the VLM is required to choose between {A black bike rests against a brown bed.} and {A black bike hangs from a brown bed.}. Similarly, in the bedroom context (observed by the VLM, but hidden from the LLM that produced the "hangs from" negative), this might be an easy choice for a VLM.

# F   SugarCrepe Partitions

Our ConMe benchmark utilizes the partitions provided by SugarCrepe [17] dataset, which consists of 919 total images[6] – 333 from the *Replace-Att* partition, 333 from *Replace-Object*, 253 from *Replace-Relation*. SugarCrepe proposes to modify the positive caption of an image by either replacing, swapping, or adding atomic concepts – which are demonstrated through different dataset partitions – in order to confuse the VLMs. To avoid language errors, SugarCrepe employs an LLM for the atomic concept manipulation and follows the manipulation by LLM-based de-biasing (ensuring that the LLM has no bias towards the augmented or the original text), yet only on the text side, disregarding the image context. On the contrary, in our work, we focus on providing image context in addition to textual context, by employing a combination of different VLMs, rather than LLMs, to generate new questions and answer options.

Below we include a summary and description of these three partitions from the baseline SugarCrepe dataset, to provide additional context on the original structure:

- *Replace-Attribute* forms a negative by replacing the attributes describing object characteristics. As an example, for an image taken on the ground, two text options are: {Several vehicles providing ground transportation are shown in the photo: streetcar, tour bus, classic car, and family cars.} and {Several vehicles providing aerial transportation are shown in the photo: helicopter, hot air balloon, small plane, and glider.}. We observe, that the negative was generated by the LLM without any image context. Hence, despite the linguistic correctness, it is unlikely a hard negative for a VLM provided with the image context of a ground.

---

[6]sourced from the MS-COCO [57] validation set

| Model | replace-att | | | replace-obj | | | replace-rel | | |
|---|---|---|---|---|---|---|---|---|---|
| | Generate | Perplexity | VQAScore | Generate | Perplexity | VQAScore | Generate | Perplexity | VQAScore |
| LLaVA 1.5-7b | 84.6 | 84.9 | 86.0 | 95.6 | 95.7 | 94.5 | 95.0 | 86.0 | 76.0 |
| InstructBLIP Flan-T5 | 88.7 | 88.7 | 92.3 | 97.2 | 97.1 | 97.0 | 88.4 | 88.7 | 85.2 |

Table 4: Comparison of accuracy (%) performance using different evaluation mode metrics on baseline SugarCrepe partitions.

Prompt for Conversation Message 1 (User Mode) to GPT-4V API

> You are a helpful AI visual assistant who can analyze images. Please describe this image in as much detail as possible. For all the details you are confident about, include everything you see, and be as specific as possible, such as describing objects, attributes, locations, lighting…

Figure 5: Step 1 Prompt

Prompt to Pipeline VLMs

> You are a helpful AI visual assistant who can analyze images. Please describe this image in as much detail as possible. For all the details you are confident about, include everything you see, and be as specific as possible, such as describing objects, attributes, locations, lighting…

Figure 6: Step 2 Prompt

- *Replace-Object* refers to negative generation via replacing the object (noun) in the positive caption. For example, given an image of a teddy bear next to some boxes in a room, a VLM is asked to choose between the *positive* {A big teddy bear sitting next to some boxes.} and the *negative* {A big car sitting next to some boxes.}. Even though the negative is grammatically correct and potentially unbiased given the partial context (a room is not mentioned in the positive text), we would not expect a car to sit next to the boxes in a room (though it might happen near the side of the road). As follows, it is unlikely that a modern VLM would be confused, as it can complete the missing details (the room) from the image and infer the unlikelihood of a car there based on the image context.

- *Replace-Relation* replaces a word describing a spatial relation between objects in a caption to form the negative. For example, given an image taken in a bedroom, the VLM is required to choose between {A black bike rests against a brown bed.} and {A black bike hangs from a brown bed.}. Similarly, in the bedroom context (observed by the VLM, but hidden from the LLM that produced the "hangs from" negative), this might be an easy choice for a VLM.

Prompt for Conversation Message 2 (Assistant Mode) to GPT-4V API

*<insert GPT-4V response from step 1>*

Prompt for Conversation Message 3 (User Mode) to GPT-4V API

The following vision-language models generated their own respective descriptions for the provided image:
LLaVA 1.6: *<insert description from step 2>*
LLaVA 1.5: *<insert description from step 2>*
InstructBLIP Flan-T5: *<insert description from step 2>*
InstructBLIP Vicuna-7b: *<insert description from step 2>*

Compositional reasoning defines the understanding of attributes, relations, and word order significance. A good vision-language model should be able to accurately answer composition reasoning questions about an image. Your task is to fool a vision-language model by generating challenging compositional reasoning questions about an image.

Given the description you generated and the descriptions these vision-language models generated, generate 10 challenging compositional reasoning questions which these models would incorrectly answer. Only create questions based on details captured in your description but lacking from the other vision-language models' descriptions. For each question, include the following:
- A compositional reasoning question
- A correct answer
- 5 hard negative options

Each negative option should differ only subtly from the correct answer but still be clearly incorrect given the image and question. The goal is for a vision-language model to choose the negative option over the positive option when asked to answer the question in binary multiple choice format. Only include questions you are confident in your answer for. Format your response as a string in the format [{"q": <question>, "a": <correct answer>, "n1": <negative option 1>, "n2": <negative option 2>…}].

Figure 7: Step 3 Prompts

Generate Inference Prompt to Pipeline VLMs Except InstructBLIP Vicuna 7B

*<insert GPT-4V generated question from step 3>*?
A. *<insert txt option generated from step 3>*.
B. *<insert txt option generated from step 3>*.
Answer with the option's letter from the given choices directly. Answer:

Generate Inference Prompt to InstructBLIP Vicuna 7B Only

*<insert GPT-4V generated question from step 3>*? Answer with the option's letter, A or B, from the given choices below.
A. *<insert txt option generated from step 3>*.
B. *<insert txt option generated from step 3>*.
Answer:

Figure 8: Step 4 Generation Inference Mode Prompts

Prompt to Pipeline VLMs

*<insert GPT-4V generated question from step 3>?*

Figure 9: Step 5 Prompt

Prompt for Conversation Message 4 (Assistant Mode) to GPT-4V API

*<insert GPT-4V response from step 3>*

Prompt for Conversation Message 5 (User Mode) to GPT-4V API

Given the image and these questions, the following vision-language models generated their own respective answers, listed in order of the questions:
LLaVA 1.6-7b: [*<insert answer to q1>*, *<insert answer to q2>*, … *<insert answer to q10>*]
LLaVA 1.5-7b: [*<insert answer to q1 >*, *<insert answer to q2>*, … *<insert answer to q10>*]
InstructBLIP Flan-T5: [*<insert answer to q1 >*, *<insert answer to q2>*, … *<insert answer to q10>*]
InstructBLIP Vicuna-7b: [*<insert answer to q1 >*, *<insert answer to q2>*, … *<insert answer to q10>*]

Generate 10 new sets of question/answer/negatives which these vision-language models would find even more challenging.

Figure 10: Step 6 Prompts

Generate Inference Prompt to Pipeline VLMs Except InstructBLIP Vicuna 7B

*<insert GPT-4V generated question from step 6>*?
A. *<insert txt option generated from step 6>*.
B. *<insert txt option generated from step 6>*.
Answer with the option's letter from the given choices directly. Answer:

Generate Inference Prompt to InstructBLIP Vicuna 7B Only

*<insert GPT-4V generated question from step 6>*? Answer with the option's letter, A or B, from the given choices below.
A. *<insert txt option generated from step 6>*.
B. *<insert txt option generated from step 6>*.
Answer:

Figure 11: Step 7 Generation Inference Mode Prompts

Perplexity Inference Prompt to Pipeline VLMs

*<insert GPT-4V generated question from step 3>*?
A. *<insert txt option generated from step 3>*.
B. *<insert txt option generated from step 3>*.
Answer with the option's letter from the given choices directly. Answer:

Figure 12: Step 4 Perplexity Inference Mode Prompts

Perplexity Inference Prompt to Pipeline VLMs

*<insert GPT-4V generated question from step 6>*?
A. *<insert txt option generated from step 6>*.
B. *<insert txt option generated from step 6>*.
Answer with the option's letter from the given choices directly. Answer:

Figure 13: Step 7 Perplexity Inference Mode Prompts

Table 5: Question Formats taxonomy specification.

| Format | Definition | Example |
|---|---|---|
| Hallucination | The question asks if something is visible or not, and the answer is that it is not visible/present. | "Is there a cat in the room?" "No, there is no cat." |
| Misconception | The question asks about an attribute of an object, but that object is not present. | "What color is the cat?" "There is no cat." |
| Non-Determinable | The question asks for something that cannot be distinguished. | "Is the cat in motion?" "I cannot tell." |
| Selective | Any other question, asking about an image detail perceived by GPT-4V as unseen by other models during our proposed ConMe curation conversation. | "What specific accessory does the person have around their neck and lower face region? A Scarf or Goggles?" "A Scarf" |

Table 6: Error Categories taxonomy specification.

| Topic | Definition | Example |
|---|---|---|
| Attention | The question asks about the attention of a person or object. | "Which direction is the cat looking?" "The cat is looking out the window." |
| Attribute | The question asks about the presence or visibility of an attribute of an object. | "Does the cat have white whiskers?" "No, the cat has black whiskers." |
| Behavior | The question asks about action or behavior. | "Is the cat moving around?" "No, the cat is sleeping." |
| Clothing | The question asks about what is being worn. | "Is the cat wearing a hat?" "No, the cat is not wearing a hat." |
| Color | The question asks about the color of an object. | "What color is the cat?" "The cat is black." |
| Count | The question asks about the number of objects. | "How many cats are there?" "There are two cats." |
| Emotion | The question asks an opinion of what is observed. | "What makes this room cozy?" "The fireplace makes the room cozy." |
| Lighting | The question asks about the lighting or direction of the light. | "Is the cat's shadow sharp?" "No, the shadow is diffused." |
| Proximity | The question asks about the spatial relation between two objects. | "Is the cat near the window?" "Yes, the cat is near the window." |
| Scene | The question asks about the location of the scene. | "Is this indoor or outdoor?" "This is indoor." |

You are an insightful assistant, for the question/answer pair provided by the user, pick a question format and question topic from the list below:
Question Format:
- hallucination: the question asks if something is visible or not, and the answer is NO, or that it is not visible/present (e.g. "Is there a cat in the room?" "No, there is no cat in the room.")
- misconception: the question asks about an attribute of an object, but that object is not present (e.g. "What color is the cat?" "There is no cat.")
- non-determinable: the question asks for something that cannot be distinguished (e.g. Is the cat in motion? "I cannot tell." OR "It is unclear.")
- selective: any other questions that do not fall into the above categories
Question Topics:
- lighting: the question asks about the lighting or direction of the light (e.g. "Is the cat's shadow sharp?" "No, the shadow is diffused.")
- clothing: the question asks about an what is being worn (e.g. "Is the cat wearing a hat?" "No, the cat is not wearing a hat.")
- attribute: the question asks about the presence or visibility of an attribute of an object (e.g. "Does the cat have white whiskers?" "No, the cat has black whiskers.")
- emotion: the question asks an opinion of what is observed (e.g. "What makes this room cozy?" "The fireplace makes the room cozy.")
- attention: the question asks about the attention of a person or object (e.g. "Which direction is the cat looking?" "The cat is looking out the window.")
- color: the question asks about the color of an object (e.g. "What color is the cat?" "The cat is black.")
- scene: the question asks about the location of the scene (e.g. "Is this indoor or outdoor?" "This is indoor.")
- count: the question asks about the number of objects (e.g. "How many cats are there?" "There are two cats.")
- behavior: the question asks about action or behavior (e.g. "Is the moving around?" "No, the cat is sleeping.")
- proximity: the question asks about the spatial relation between two objects (e.g. "Is the cat near the window?" "Yes, the cat is near the window.")
Do not confuse formats with topics.
Respond with a JSON object with the following format:
{
    "question_format": "format",
    "question_topic": "topic"
}

Figure 14: The complete prompt to the Llama-3 [63] used to classify different questions in the ConMe dataset according to the question format and question topic for analysis of VLM errors.

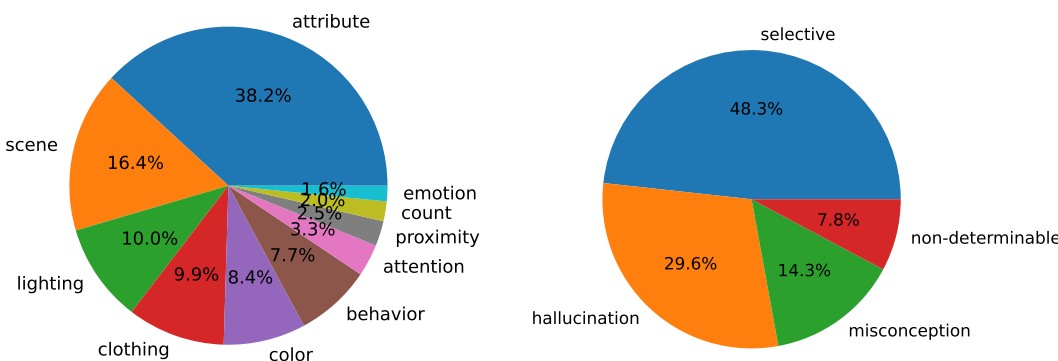

Figure 15: Percentage of samples belonging to different categories classified by Llama-3, according to CR Q/A topic (left) and CR Q/A format (right).

| Image | Baseline | Pipeline |
|---|---|---|
| 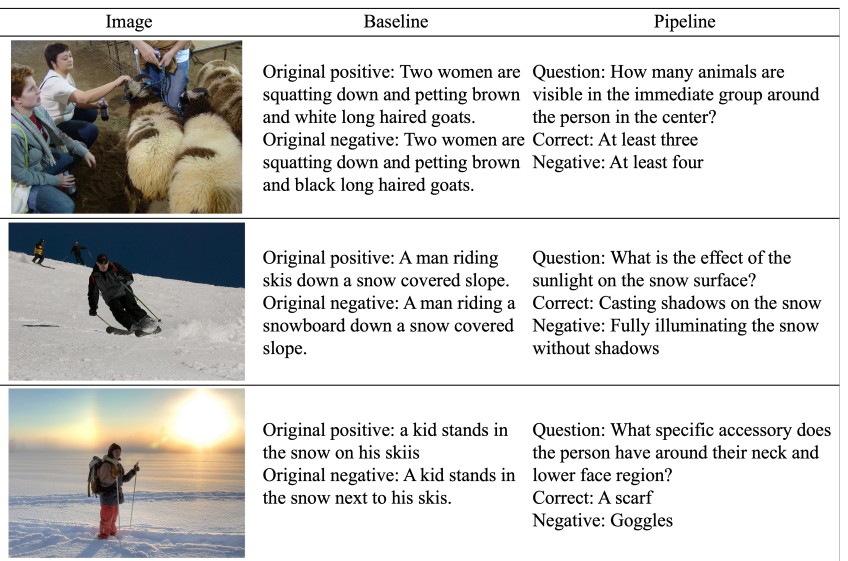 | Original positive: Two women are squatting down and petting brown and white long haired goats. Original negative: Two women are squatting down and petting brown and black long haired goats. | Question: How many animals are visible in the immediate group around the person in the center? Correct: At least three Negative: At least four |
| | Original positive: A man riding skis down a snow covered slope. Original negative: A man riding a snowboard down a snow covered slope. | Question: What is the effect of the sunlight on the snow surface? Correct: Casting shadows on the snow Negative: Fully illuminating the snow without shadows |
| | Original positive: a kid stands in the snow on his skiis Original negative: A kid stands in the snow next to his skis. | Question: What specific accessory does the person have around their neck and lower face region? Correct: A scarf Negative: Goggles |

Figure 16: Randomly chosen qualitative examples from the SugarCrepe and the proposed ConMe benchmark.

