# OpenReview forum: "ConMe: Rethinking Evaluation of Compositional Reasoning for Modern VLMs"
_NeurIPS.cc/2024/Datasets_and_Benchmarks_Track — NeurIPS 2024 Track Datasets and Benchmarks Poster_

### Official Review · Reviewer_pFDo · 2024-07-12
**Some significant doubts should be addressed**

**Rating:** 6
**Confidence:** 3
**Correctness:** Pls see above.
**Clarity:** Fairly well written.

**Review:**

**Positives:**

**P1:** Good motivation (to create datasets that push the boundaries of CR).

**P2:** The dataset "works", in the sense that from Table 2, performance on ConMe is worse than on SugarCrepe. [But see N3 below.]


**Negatives:**

**N1a:** Would be good to include one actual example "conversation" among the VLMs to get a better sense of what really happens using this approach.

**N1b:** The pipeline seems a little bit arbitrary. Is there a more principled or well-justified line of reasoning for this "conversation" approach?

**N2:** Is this "conversation" approach significantly different **and** better than just "somehow" (there are many ways) producing a large set of CR examples beforehand, and then filtering out the examples that all 4-5 VLMs (5, if GPT-4V is included) get correct? [I know this second step is also what ConMe does; L166.]

**N3.1:** From Table 2, the "real" results are really the 12.3% and 15.4% drop on InternLM and Idefics, since they were not involved in the process of creating ConMe. It is very telling that the performance drops of the LLaVA and InstructBLIP were much larger. So, if one were to be completely objective, for all that effort, ConMe is "just" 12.3% to 15.4% harder?

**N3.2:** I have a **strong objection** to the abstract and introduction only stating the "up to 33% decrease". Minimally, the mean performance drop of 23.3% should be reported instead. But ideally, the reported number should be related to the 12.3% and 15.4% on the "unseen"  VLMs (e.g. take the mean or report both numbers).


**Questions:**

**Q1:** Why are 2 versions of 2 models (LLaVA and InstructBLIP) used? Why not use 4 completely different models?

**Q2:** L146: What happens if the GPT-4V description is wrong?

**Q3:** Why is a second iteration needed, or this is purely an empirically-based decision (e.g. see Table 3)? (i.e. 2 iterations works better than 1 or 3). The explanation on L176 seems like a post-hoc one.

**Q4:** From Table 3, it looks like Iteration 2 often "makes things worse", i.e. leads to higher accuracy? As just one example, looking at the "Generate" accuracy for "Iteration 1 + Filtering" versus "Iteration 2 + Filtering", both InstructBLIP models have higher accuracy after Iteration 2, which is a bad thing?


**Minor (for consideration only, no need to respond):**

**M1.**  L39: "failure" instead of "fail"?

**Strengths:**

Pls see above.

**Additional Feedback:**

Nil.

**Documentation:**

Based on a brief look at the websites for the dataset and code, as well as details provided in the paper and supplementary material, reproducibility seems sufficiently supported.

**Ethics:**

No major concerns at this juncture.

**Limitations:**

Pls see above.

**Opportunities For Improvement:**

Pls see above.

**Relation To Prior Work:**

Sufficiently clear how the work differs from previous work.

**Summary And Contributions:**

This paper proposes a methodology/framework and dataset for Compositional Reasoning (CR) that claims to produce a new dataset (ConMe) that current SOTA VLMs do significantly worse on, compared to the latest datasets (e.g. SugarCrepe). This shows that SOTA VLMs are still far from "solving" the CR challenge.

---

> ### Author Rebuttal · Authors · 2024-08-16
>
> Thank you for the time and effort spent in reviewing our paper. In the following, we respond to the questions raised in the review.
>
> **One complete example conversation (N1a).** Thank you for pointing it out. We have included one conversation in the *global_response.pdf* and will also include this in the updated manuscript.
>
> **Reasoning behind the proposed "conversation" approach (N1b & Q3).** Our proposed methodology provides a pathway for different VLMs to interact with each other through a common language interface. The intuition behind employing a multi-stage conversation methodology between the GPT-4V and several downstream VLMs is that we want to expose the weaknesses of these VLMs to the GPT-4V in an interactive manner. In multiple calls to the GPT-4V, the model can consume additional context, regarding the perception abilities of the other models and can self-reflect upon the generated negatives in the preceding step.
> Furthermore, this iterative scheme of question generation and filtering ends up with ConMe, where the GPT4V can generate questions that expose its weaknesses. In summary, the idea of employing conversation between the models lets these models collaborate and iteratively find challenging examples.
>
> **Differences in the proposed approach from generating a large amount of CR examples and then filtering (N2).** The main difference between our *conversation-based* approach and generating a large amount of CR examples and then filtering is that there will be no concept of 'interaction' between the different models, and they will not be able to *collaboratively* explore the weaknesses of each other, which is one of the cornerstones of our proposed methodology.
>
> To answer the reviewer's question quantitatively: we tried the approach of generating a large amount of CR questions and then filtering and this resulted in a dataset with very few samples (~200 vs. ~24k in ConMe) and also resulted in higher error rates after manual verification (much higher error rate of ~40%).
> The lower number of samples is because GPT4-V does not know what the other VLMs can perceive (because of lack of conversation) and thus, the resulting CR questions are *easier* for the models to answer and thus, most of the generated CR questions are filtered because of that. Furthermore, the higher error rates stem from the fact that the samples which passed the filtering stage, are likely erroneous.
> On the other hand, for ConMe, since the GPT4-V already knows the perception abilities of the other VLMs (due to the conversation-based methodology), thus, it can exploit these abilities for *more challenging* CR questions which are also less erroneous due to the prior knowledge GPT4-V has from all the other VLMs, resulting in lower error-rates.
>
>
> **Performance drops and objections on writing (N3.1 & N3.2).** The main motivation behind our work is to present a method that can collaboratively find the weaknesses of any downstream VLMs and then later we also provide a toolbox to analyze those weaknesses through our taxonomy analysis. As pointed out by the reviewer, the performance drop on the models that are *involved* in this *conversation-based methodology* is indeed higher. Furthermore, to show the generalization of our proposed benchmark, we also test it on unseen VLMs, where the performance drops are lower. However, we believe that it is a slight injustice to our proposed method and benchmark to quantify its *hardness* based on just the performance drops on unseen VLMs because our method in principle can be extended to any number of VLMs (including the two *unseen* VLMs used in this work) and find their weaknesses as well, which also makes our proposed method and the benchmark more adaptable for future VLMs.
>
> We also want to point out that we will take into account the reviewer's suggestion to update our manuscript in the next iteration and will report the performance drops explicitly for *seen* and *unseen* models.
>
>
> **Different types of models used (Q1).** In our work, we have employed 4 open-source VLMs: LLaVA 1.5-7b, LLaVA 1.6-7b, InstructBLIP Flan-T5 and InstructBLIP Vicuna-7b. These models are widely used by the computer vision research community and have been chosen by us because of the inherent differences in their model design and performance.
>
> - *LLaVa-1.5 and LLaVa-1.6* have huge differences in their performances and visual reasoning capabilities. In LLaVa-1.6, the authors have incorporated *anyres* training and inference. Specifically, they have increased the input resolution to 4x more pixels. This allows the model to grasp more visual details. It supports three aspect ratios, up to 672x672, 336x1344, 1344x336 resolution.
> - *InstructBLIP Flan-T5 and InstructBLIP Vicuna-7b* have inherently different architectures because of the type of language model they use and that also results in different performance and reasoning abilities, as reported in the original InstructBLIP paper. Flan-T5 is an encoder-decoder LLM, advantageous for structured tasks where the input-output relationship is clear and following instructions is crucial. It performs well on a broad range of NLP tasks, especially those requiring a fine-grained understanding of task descriptions. Vicuna is more efficient for text generation tasks, particularly in scenarios where inference speed is important or where the task is open-ended and creative. It may require careful prompt engineering but can be highly effective in various contexts.
>
> In summary, the choice is motivated by the difference in the visual reasoning capabilities of these models and widespread adaptability. Furthermore, we would also like to point out that our proposed pipeline is highly adaptive (and general) and can incorporate any VLMs that are also developed in the future.

---

> > ### Author Rebuttal · Authors · 2024-08-16
> >
> > **Erroneous descriptions from GPT4-V (Q2).** GPT-4V has been widely adopted by the computer vision community for the task of visual reasoning and is also considered as one of the more capable VLMs to date. It has been used by [1, 2, 3, 4, 5, 6] (and many others) to curate datasets in multiple different domains. However, as pointed out by the reviewer (and we agree) GPT-4V is still a VLM and is prone to make mistakes and this is the reason we also point this out as a limitation of our work.
> >
> > To account for such mistakes, we did manual verification and error analysis and released a manually verified subset of 1000 samples. The results indicate that the errors (which might be due to erroneous GPT4-V descriptions) are overall uniformly distributed and largely independent from mistakes made by
> > the VLMs which results in negligible performance fluctuation between performance predicted by (unfiltered) ConMe pipeline data and 1000 human-validated CR QA from ConMe, as reported in Table 2.
> >
> >
> > **InstructBLIP results after filtering of the second stage (Q4).** In stage 2, even if InstrcutBLIP is used in the generation pipeline, the pipeline could result in samples that are not necessarily more difficult for InstructBLIP, but generally more difficult for the other VLMs (which are or are not included in the generation pipeline). This is also the reason why we employ several VLMs - to obtain more difficult samples that generalize to other multimodal models.
> >
> > [1] mPLUG-DocOwl 1.5: Unified Structure Learning for OCR-free Document Understanding
> >
> > [2] Cambrian-1: A Fully Open, Vision-Centric Exploration of Multimodal LLMs
> >
> > [3] Visual Instruction Tuning
> >
> > [4] ShareGPT4V: Improving Large Multi-Modal Models with Better Captions
> >
> > [5] ShareGPT4Video: Improving Video Understanding and Generation with Better Captions
> >
> > [6] To See is to Believe: Prompting GPT-4V for Better Visual Instruction Tuning

---

> > > ### Author Response · Authors · 2024-08-21
> > > **Author-Reviewer Discussion Period**
> > >
> > > Dear Reviewer pFDo,
> > >
> > > We hope this message finds you well. We apologize for reaching out to you. We are extremely grateful for your valuable comments on our work. As the author-reviewer discussion phase is in progress, we want to know if our rebuttal has effectively addressed your concerns. Your feedback and response are very important to us. If you have any questions or need further clarification about our work, please do not hesitate to let us know immediately. We greatly appreciate your understanding and support.
> > >
> > > Thank you very much.

---

> ### Comment · Reviewer_pFDo · 2024-08-26
>
> Thanks to the authors for a thorough rebuttal. Most of the less-subjective issues have been addressed satisfactorily, with the assumption that any promised changes will be made. As such, overall I find the paper to be of acceptable quality, and I have raised my score from 4 to 6.

---

> > ### Author Response · Authors · 2024-08-26
> > **Thank you!**
> >
> > Dear Reviewer pFDo,
> >
> > Thank you for your time and efforts. We are extremely thankful to you for acknowledging our rebuttal and updating your score. Of course, all changes will be incorporated in the updated manuscript, as promised.

---

### Official Review · Reviewer_b3yx · 2024-07-24
**Interesting method for generating dataset for benchmarking composition reasoning in VLMs but the paper lacks qualitative results and sufficient explanation for the results**

**Rating:** 6
**Confidence:** 4

**Review:**

The paper presents a new benchmark for compositional reasoning in vision language models called ConMe. The paper is largely well written and the method used for generating the dataset used in benchmark is also interesting. The benchmark results suggest that even the strongest VLMs have not perfectly solved visual understanding and might help us advance the SOTA.

Having said that, I’m kind of torn on this paper. While the paper includes a nice dataset for evaluation of SOTA VLMs and a novel method for generating more/harder CR questions given a set of images, there are a few concerns including over-reliance of the method on one specific VLM, lack of qualitative results on both VLM results and the difference between generated and prior datasets which makes it difficult to accept the paper outright. The exposition of the paper can also be improved significantly. Therefore, I’m currently rating it as borderline leaning towards rejection but would be willing to increase the rating if the authors can sufficiently address the concerns and other reviewers do not raise other major concerns that I might have overlooked.

Please refer to the relevant sections below for detailed strengths and weaknesses.

**Updated score after rebuttal. Please see the comment below for details.**

**Strengths:**

- The paper tackles a very interesting and relevant task of compositional reasoning in vision language models which aims at improving general visual understanding in SOTA models.
- The paper proposes a simple, iterative and scalable method to generate new / more difficult CR questions given a set of images based on SOTA VLMs i.e. given access to stronger models, the method could generate harder questions.
- As suggested by results of GPT4V, even the “orchestrator” model can generate questions that can fool itself and the method could potentially also be used to generate data to be used to improve these models on the benchmark.

**Additional Feedback:**

N/A

**Clarity:**

The paper is mostly well written with exception of a few parts as outlined in “opportunities for improvements” above.

**Correctness:**

Yes, the dataset generated is mostly correct.

However, one worrying factor is lack of control over what questions are generated. GPT4V is strong but at the end of the day it still is an imperfect VLM – I worry what proportion of the questions generated are related to mis-classifications due to GPT4V.

**Documentation:**

The paper includes links for both the data on huggingface and evaluation code on github.

**Limitations:**

The authors include a limitations section that sufficiently addresses the limitations/concerns.

**Opportunities For Improvement:**

- **The method is too reliant on GPT4V.**
The method involves GPT4V looking at an image and describing it in natural language which is used as a seed for generating questions in later iterations. This means that anything that is missed by GPT4V will not end up in the questions and more worryingly sometimes GPT4V hallucinations might also end up in the questions.

- **Does SugarCrepe really contain irrelevant negative examples?** The paper claims but does not sufficiently prove / explains the problem with SugarCrepe dataset. For example, the paper claims that negative descriptions in SugarCrepe might be unlikely based on the scene observed in the image but does not provide any relevant example from the dataset. The only 3 examples shown in Figure 14 are in fact highly relevant to the paired images.

- **Lack of qualitative examples.**
While the results are fine, I am unable to form any intuition of the type of questions ConMe includes which SugarCrepe or any other dataset does not include and that makes it difficult for these models.
It would also have helped much more to see the qualitative examples than the aggregated “error taxonomy analysis” in Section 5.2.
A side note – I’m not really convinced the section is really relevant to the benchmark at hand. Moreover, it is also the most difficult to understand due to weird phrasing and markedly lower quality of writing than the rest of the paper which really hurts the overall exposition.

- **Discrepancy of InterLM and GPT4V results.** InternLM-XComposer2-VL-7b seems to be performing even better than GPT4V on both SugarCrepe and the manual subset of the benchmark which is a bit suspect, especially given the fact that the questions were generated from descriptions of GPT4V. Again, it would have been very helpful to see qualitatively what kind of questions does it get right and GPT4V gets wrong.

- **On a high level, what do the results suggest?**
I feel “compositional reasoning” has been used in a weird way in this sub-field (not a problem of just this paper). A very small sample of questions that I can see in Figure 14, suggest that these probably only evaluate the models ability to pick up on fine-grained information in an image. What is so “compositional” about it? Would having a better vision head just solve this issue or is there a more fundamental problem with these models?

**Relation To Prior Work:**

The paper clearly discusses previous work and sufficiently addresses the differences.

**Summary And Contributions:**

The paper highlights problems with a prior compositional reasoning benchmark SugarCrepe and proposes an “automatic” fix based on GPT4V.

Concretely, most prior compositional reasoning benchmarks pose the problem as a text/caption retrieval problem where a model is presented with an image, a positive text description and a negative description and evaluated on its ability to select positive text description. This paper posits that the negative descriptions in the prior benchmarks are too easy to discern due to either flaws in language or irrelevance to the image. Specifically, they focus on the SugarCrepe benchmark where the negatives are generated using “blind”-LLMs which produce “easy” negatives as many descriptions have no relevance to the image. The paper fixes this problem by using strong VLMs to generate those negatives in an iterative fashion.

The paper builds on top of SugarCrepe and generates a much larger number of questions compared to the original version. The resulting benchmark is called ConMe. The paper then benchmarks multiple VLMs and shows that all of them struggle to perform as well on ConMe as on SugarCrepe.

---

> ### Author Rebuttal · Authors · 2024-08-16
>
> Thank you for the time and effort spent in reviewing our paper. In the following, we respond to the questions raised in the review.
>
> **Reliance on GPT4-V (Q2).** GPT-4V has been widely adopted by the computer vision community for the task of visual reasoning and is also considered one of the more capable VLMs to date -- which is also a reason why we employ it in our methodology. It has been used by [1, 2, 3] (and many others) to curate datasets in multiple different domains. However, we agree that GPT-4V is still a VLM and is prone to make mistakes (can hallucinate and might even miss details in an image) and this is the reason we also point this out as a limitation of our work.
>
> To account for such mistakes, we did manual verification and error analysis and released a manually verified subset of 1000 samples. The results indicate that the errors (which might be due to erroneous GPT4-V descriptions) are overall uniformly distributed and largely independent from mistakes made by
> the VLMs which results in negligible performance fluctuation between performance predicted by (unfiltered) ConMe pipeline data and 1000 human-validated CR QA from ConMe, as reported in Table 2 (main manuscript).
>
> **Irrelevant examples in the SugarCrepe dataset.** We analyzed samples in the SugarCrepe dataset and found many irrelevant positive and negative pairs. We provide a few examples in the *global_response.pdf* (Figure 2), with a brief rationale about why the positive or the negative is irrelevant to the image. The irrelevance of these examples stems from the fact that during the creation, SugarCrepe does not have access to the images and that leads to hallucinations. For example, in the third example (in Figure 2) the positive and negative is about a *plant flying/swaying* in the air. However, there is no plant present in the image, but rather a plane is flying. We will further expand on this and include more such examples in the updated manuscript.
>
> **How are generated questions more difficult in ConMe?** We manually compare a few questions and the associated negative/positive pairs from ConMe with the SugarCrepe dataset and provide three such examples in Figure 3 of the *global_response.pdf*.  We observe that our generated questions are more challenging. For example, in the first image, our generated question is about the parking position of the vehicle, which requires an understanding of spatial relationships and proximity in the scene concerning the parking lanes, whereas, the SugarCrepe positive/negatives are easier since the image has many giveaways, like sky and trees, which makes it very easy for the model to reason that the parking lot is *outdoor*. Similarly, in the third image, our generated question requires multi-hop reasoning about the type of agriculture *implied* in the scene. On the other hand, the positives and negatives from the original SugarCrepe dataset only require the model to know if the land is *rocky* or *grassy*, which is very straightforward to answer from the given image, that the land is *grassy*. We will further add more qualitative examples in the updated version of the manuscript.
>
>
> **InterLM-Xcomposer2-VL performs better than GPT4V?** InterLM-Xcomposer2-VL performs slightly better on ConMe and the SugarCrepe dataset. In the original paper of InterLM-Xcomposer2-VL (Table 3), we observe that the InternLM-XComposer2 can outperform GPT4V on some benchmarks, thus, it can be the case that this model is slightly stronger on compositional reasoning than the GPT4-V.
> We also want to point out that we again verified the evaluation pipeline for the model and did not find any bug, thus, the evaluations are correct. Furthermore, due to space constraints in the allowed 1-page pdf, at this point, we are unable to provide a qualitative example, but we will add the examples where GPT4-V is failing and InternLM succeeds.
>
> **Concept of Compositionality Reasoning and ways to solve it.** Compositional in this context means not only noticing fine details but also reasoning about combinations of those details to get an answer. From Figure 14 (image 1) counting the sheep indeed requires reasoning about *compositionality*. For Figure 14 (image 2) we need to cross-reference the snow with the likely position of the sun, the position of the skiers and the likelihood of shadows in respective darker pixels on the snow, which also requires reasoning about *compositionality*. Similarly, in Figure 14 (image 3), the object around the neck needs the location of the next and recognition of the small object. However, we also believe that fine-grained visual understanding is indeed a part of CR.
>
> Many present-day vision-language models are built upon the vision encoder from CLIP which has shown to inherit a bag-of-words behavior. Recent research [4] has shown that CLIP models exhibit an ‘object bias’ partially due to the contrastive text-to-image loss used in their pre-training. For example, the popular CLIP (contrastive) loss is computed over a random batch of text-image pairs sampled from a large-scale and diverse VL dataset with the chance of two images in the same batch containing the same set of objects being very low. For such a loss, representing just a ’bag of objects’ in each image or text is sufficient for matching the corresponding pairs. Thus, it can be the case that the CLIP vision encoder inherits these biases and transfers them to the modern encoder-decoder VLMs (e.g., [3]). One possible way to improve this behavior is to replace the contrastively pre-trained vision backbone with a vision-only pre-trained backbone like DINO-V2.
> Recently, Tong et. al. [5] showed that it is indeed possible to get some improvements in visual reasoning by replacing the vision backbones in VLMs with DINO-V2. Thus, in the future, such solutions can be explored more and they might provide further improvements, however, reasoning and cross-referencing details in a VLM might always remain on the LLM side.

---

> > ### Author Rebuttal · Authors · 2024-08-16
> >
> > [1] Visual Instruction Tuning
> >
> > [2] ShareGPT4V: Improving Large Multi-Modal Models with Better Captions
> >
> > [3] To See is to Believe: Prompting GPT-4V for Better Visual Instruction Tuning
> >
> > [4] Teaching Structured Vision & Language Concepts to Vision & Language Models
> >
> > [5] Eyes Wide Shut? Exploring the Visual Shortcomings of Multimodal LLMs

---

> > > ### Author Response · Authors · 2024-08-21
> > > **Author-Reviewer Discussion Period**
> > >
> > > Dear Reviewer b3yx,
> > >
> > > We hope this message finds you well. We apologize for reaching out to you. We are extremely grateful for your valuable comments on our work. As the author-reviewer discussion phase is in progress, we want to know if our rebuttal has effectively addressed your concerns. Your feedback and response are very important to us. If you have any questions or need further clarification about our work, please do not hesitate to let us know immediately. We greatly appreciate your understanding and support.
> > >
> > > Thank you very much.

---

> > > > ### Comment · Reviewer_b3yx · 2024-08-25
> > > >
> > > > I thank the authors for a really good rebuttal.
> > > >
> > > > Looking at the responses and qualitative examples in the attached pdf, I'm a bit more positive about the paper and have updated the score to 6 (from 5). I specially appreciate the examples from SugarCrepe dataset as it makes the difference a lot more clear and would encourage the authors to include more of such examples in the updated version of the paper including qualitative examples of the performance of the VLMs too.
> > > >
> > > > Additionally, I would echo the concern from reviewer pFDo in the bullet point N3.2 which I had noticed but failed to include the review and would encourage the authors to also make relevant changes to avoid exaggerating performance differences in the Abstract and Introduction.

---

> > > > > ### Author Response · Authors · 2024-08-26
> > > > > **Thank you!**
> > > > >
> > > > > Dear Reviewer b3yx,
> > > > >
> > > > > Thank you for acknowledging our rebuttal and increasing the score. We will make sure to include the changes in the updated version of the manuscript.

---

### Official Review · Reviewer_GWAy · 2024-07-27
**A good benchmark with potential for significant impact on the development of VLMs**

**Rating:** 9
**Confidence:** 4
**Clarity:** Yes

**Review:**

This is a high-quality submission. It proposes a benchmark with a potential for significant R&D impact. The paper is written well, and the experimental details and results are satisfactory. The approach is sound and reasonable, and more importantly, the results are compelling.

There are no significant ‘cons’ that may weigh against accepting the paper. The paper could be improved by adding some more analysis and discussions, and by fixing some clarity issues (mentioned below in the detailed review).

More details for both the pros and the cons are provided below.

**Strengths:**

- **Relevance**: The paper is motivated well and addresses an important problem of interest to the vision and language community, and to the NeurIPS D&B community in general.
- **Clarity**: The quality of writing is good; the exposition is clear and easy to understand.
- **Originality/ Novelty**: The paper has good novelty.
  - A novel computational reasoning benchmark, ConMe, is proposed that challenges modern VLMs.
  - The proposed benchmark creation pipeline has good novelty. It uses GPT-4V (in a primary role) and other VLMs to automatically and collaboratively create a hard dataset that the VLMs (and even GPT-4V) find hard to perform computational reasoning on.
- **Significance**: ConMe is a moderately sized (significantly larger than SugarCrepe) and compelling benchmark. It provisions a subset of images from the SugarCrepe dataset with a corresponding Q&A set which leads to a significant performance drop in the performance of SOTA algorithms of the LLaVA, InstructBLIP, InternLM, Idefics family (-15% to -33%) and even for GPT-4V (-11%) and is expected to have a significant impact on the development of VLMs with a better CR performance.
- **Experimental Validation**:  Adequate. Results on a manually curated subset of 1000 samples supports the results on the automatic curation and Q&A generation pipeline. Ablation studies demonstrate the value from different parts of the pipeline.
- **Reproducibility**: Results should be reproducible as an official gitHub code repository has been created.

**Additional Feedback:**

No additional comments.

**Correctness:**

The paper proposes the ConMe benchmark which consists of (a) subset of images curated from the SugarCrepe dataset, and (b) corresponding Q&A both curated through an automated pipeline comprising of GPT-4V and other VLMs to create a much harder dataset. The task is a forced binary choice VQA setting with binary accuracy as the metric. The experimental design seems reasonable and the evaluation rigorous. Claims seem reasonable as well (a small reservation as pointed out above).

**Documentation:**

Yes.

**Ethics:**

No concerns.

**Limitations:**

Yes. A fully automated benchmark and evaluation framework entirely using AI models (VLMs) may give erroneous evaluations which when used by the research community to improve models (to perform better on the benchmark) can lead to undesirable learning imperatives. The authors alluded to this limitation in the paper and to analyze its significance and impact, performed a comparative evaluation on a small human-verified subset and find that the errors in ConMe are small (not significant) and uniformly distributed.

**Opportunities For Improvement:**

I’m listing below some shortcomings. Addressing these will improve the quality of the presented work.
- Statement on out of distribution training examples in other CR benchmarks with respect to the text-only marginal or text-image paired data distributions (e.g. lines 6-10, …) is made multiple times in the paper and presented as a premise for the presented work. However, it is left as a conjecture without an attempt to qualitatively or quantitatively validate them.
- Discussion, illustrated by examples, of why GPT-4V may be able to create CR Q&A which it itself finds hard to accurately answer (degraded performance in Table 2) is missing and would be an important value-add.
- Examples of errors and limitations should be included: while results on a manually verified subset support the main results of the paper, using it to analyze the mistakes and error modes of the automatic generation pipeline is missing.
- (Clarity) In Figures 4 and 5, while the logic of arranging bars in a sorted order of performance is understandable, it leads to considerable difficulty in comparing across models, especially if the number of categories is larger, as in Figure 4.

**Relation To Prior Work:**

Yes. Related SOTA is adequately discussed, with subsections on recent VLMs and compositional reasoning benchmarks. The paper argues for a CR benchmark that can challenge more recent VLMs beyond the extant benchmarks like Crepe, SugarCrepe. In fact, the proposed ConMe benchmark is constructed via ‘hard-data mining’ on the SugarCrepe dataset to address the observation that current benchmarks are easily hackable by VLMs and that ‘visually blind’ models (having no access to image data) outperform modern VLMs on these benchmarks. It is shown that the performance of SOTA VLMs degrades significantly on ConMe.

**Summary And Contributions:**

The paper proposes a novel compositional reasoning benchmark called ConMe. While existing benchmarks have challenged VLMs using multimodal encoders, they are inadequate to challenge modern, promptable VLMs due to their reliance on LLM-only negative text generation pipelines. The paper proposes a more powerful data generation pipeline which curates a harder dataset from the SugarCrepe dataset using several VLMs in a novel and compelling data generation pipeline where GPT-4V is prompted to create Q&As that other VLMs will find challenging.  ConMe is shown to lead to a 33% reduction in CR performance of SOTA VLMs compared to their performance on existing benchmarks.

---

> ### Author Rebuttal · Authors · 2024-08-16
>
> Thank you for the time and effort spent in reviewing our paper. In the following, we respond to the questions raised in the review.
>
> **Out-of-distribution examples in other CR benchmarks.** Thank you for pointing it out. This can indeed help improve our paper. In Figure 2. of the *global_response.pdf* we have included qualitative examples for the SugarCrepe dataset. From the Figure, we observe that due to the lack of visual information employed by other benchmarks, the LLM can sometimes hallucinate about the details present in the images. For example, in the last image of Figure 2, there is no plant present, and certainly, a plant cannot *fly* or *sway* in the sky. Similarly, we also checked the ARO dataset and found many examples to be out of distribution. For example, there are some negatives like 'Black and yellow photograph of a tractor on the beach next
> to boats.' and 'An unstuffed bear is sitting outside an eatery.'. Please note, that there is no such thing as a *black and yellow photograph* and also *unstuffed bear* can be considered as out of distribution. We conjecture that these mistakes are due to the rule-based substitution of words for negative generation in ARO. Due to space restrictions in the allowed PDF file, we cannot upload the images for ARO, but in the updated version of the manuscript, we will add the ARO and more SugarCrepe examples with images.
>
> **Why does GPT struggle with questions it generates?** We also found it interesting that GPT4-V also struggled on the ConMe dataset. We conjecture that it is because of our interactive 'conversation-based' compositional reasoning question generation pipeline, which ends up with a complex CR dataset. In our pipeline, multiple VLM descriptions are fed to the GPT4-V as context for the generation of the CR questions, and the GPT4-V is instructed to compare these descriptions with the description it generated itself and generate the questions.
> In the end, we find that GPT4-V also struggles with the dataset it helped to curate.
> It might be because GPT4-V finds it easier to compare descriptions and generate questions based on that but struggles to answer those questions.
> For example, GPT4-V might easily generate the question "What could be the relationship between these two subjects in the image?"  as it could easily recognize objects in an image and formulate this question by statistical association.
> However, answering this question might require very deep understanding and cognitive reasoning in a deeper semantic way, which might be a difficult task for GPT4-V and hence can result in a performance drop.
> Our error analysis in Figure 3. shows that GPT struggles most with questions requiring a notion of *proximity* and *scene* understanding, both of which might require complex reasoning abilities to answer.
>
> Furthermore,  West et. al. [1] analyzed different generative models and found that the model's generation ability is not directly correlated with its understanding abilities.
> Specifically, they find that although models can
> outperform humans in generation, they consistently fall short of human capabilities in measures of understanding, showing a weaker correlation between generation and understanding performance.
> We believe that this can also be the reason why GPT4-V might not be to able to answer what it *generated*.
>
> **Error analysis on manually verified samples.** We want to point out that we performed all the error analyses on the manually verified 1000-sample subset. This is stated in line 294: "For our analysis tool, we utilized LLaMa-3 8B to categorize our human-filtered ConMe partition." Here the human-filtered ConMe partition refers to the 1000-sample subset. In the updated manuscript, we will further clarify this.
> This 1000-sample subset is used for the generation of the taxonomy of CR QA formats (Table 5) and the taxonomy of error modes (Table 6).
> The distribution of error rates w.r.t these two dimensions (for all models) are shown in Figure 3 and Figure 4 (main manuscript).
>
>
> **Clarity in figures.** Thank you for pointing it out. We will consider this suggestion and update the manuscript such that the error partitions appear at the same position for all the models (without sorting). Hopefully, that will make the figures and results easier to understand.
>
> [1] THE GENERATIVE AI PARADOX: “What It Can Create, It May Not Understand”

---

> > ### Comment · Reviewer_GWAy · 2024-08-26
> > **Acknowledgment of the rebuttal**
> >
> > Thanks for the response. This concludes the conversation from my side. All the best.

---

> > > ### Author Response · Authors · 2024-08-27
> > > **Thank you!**
> > >
> > > Dear Reviewer GWAy,
> > >
> > > We would again like to extend our gratitude for the time and effort spent in reviewing our paper and also acknowledging our rebuttal. All the new discussion points will be included in the revised manuscript.

---

### Author Rebuttal · Authors · 2024-08-16

We thank all the reviewers for their efforts to review our paper and for providing insightful feedback.

We are happy that they found our work **Novel** $(GWAy, b3yx)$, **Extensively Evaluated** $(GWAy)$, **Significant** $(GWAy, b3yx)$, and **well motivated and well written** $(GWAy, b3yx, pFDo)$.

In the attached PDF, we present more insights to gain a further understanding of our ConMe. We refer to this PDF throughout the response (to each reviewer) as *global_response.pdf*.

---

### Decision · Program_Chairs · 2024-09-26

**Decision:**

Accept (Poster)

**Comment:**

This paper proposes a data generation framework called ConMe. It works by having VLMs answer increasingly difficult questions about an image. Questions are sourced from GPT-4V and keep increasing in difficulty by asking it to generate even more difficult questions based on the answers from the previous round. This interactive setting is meant to produce data that fills a gap with existing benchmarks for evaluating compositional visual reasoning, where negative examples are typically sourced without taking the image into account. It is shown how ConMe generates questions that are much more difficult for VLMs and they generalise to evaluating VLMs that did not participate during generation. Finally, an analysis of the data generation process and failure cases of VLMs is presented.

This paper initially received mixed reviews and one very positive review, which were increased after a strong rebuttal by the authors. Reviewers are in agreement that the proposed approach is interesting and novel, and that there is a need for harder benchmark tasks (though this does need to be demonstrated quantitatively as a reviewer points out). Reviewers agree that the analysis is comprehensive and that the comparisons are valid. A number of concerns were expressed in relation to the reliance on GPT-4V, which I agree is not ideal. On the other hand, compared to having a human participate in the question generation loop, it might still end up lowering the barrier to going through this process for other datasets. What is important is that a sample of the questions and answers are verified manually, which is the case. There is also a concern about the need to go through multiple iterations of question generation, which the authors address in their rebuttal, and limited qualitative results presented, which will be added to the paper (and a preview was shown in the response). Other, more subjective issues, are concerned with the presentation of results, where the authors should paint a more accurate picture of their results. Aside from the evaluation presented and the data obtained from the laborious generation process, it is not clear whether this approach will see a lot of adaptation. In that sense, though this paper contributes an interesting dataset and paradigm, it is still borderline in my view, and my recommendation for acceptance is largely based on the positive reviews that changed their mind during the rebuttal.